# Synthesis of Graphene Quantum Dots by a Simple Hydrothermal Route Using Graphite Recycled from Spent Li-Ion Batteries

Lyane M. Darabian, Tainara L. G. Costa ⬤, Daniel F. Cipriano, Carlos W. Cremasco, Miguel A. Schettino, Jr. and Jair C. C. Freitas *⬤

Laboratory of Carbon and Ceramic Materials, Department of Physics, Federal University of Espírito Santo, Vitória 29075-910, ES, Brazil
* Correspondence: jairccfreitas@yahoo.com.br

**Abstract:** Graphene quantum dots (GQDs) are nanosized systems that combine beneficial properties typical of graphenic materials (such as chemical stability, biocompatibility and ease of preparation from low-cost precursors) with remarkable photoluminescent features. GQDs are well-known for their low cytotoxicity and for being promising candidates in applications, such as bioimaging, optoelectronics, electrochemical energy storage, sensing and catalysis, among others. This work describes a simple and low-cost synthesis of GQDs, starting from an alcoholic aqueous suspension of graphene oxide (GO) and using a hydrothermal route. GO was prepared using graphite recycled from spent Li-ion batteries, via a modified Hummers method. The GO suspension was submitted to hydrothermal treatments at different temperatures using a homemade hydrothermal reactor that allows the control of the heating program and the assessment of the internal pressure generated in the reaction. The synthesized GQDs exhibited bright blue/green luminescence under UV light; showing the success of the chosen route and opening the way for future applications of these materials in the field of optoelectronic devices.

**Keywords:** graphene quantum dots; hydrothermal synthesis; graphene oxide; graphite; recycling

## 1. Introduction

The research on carbon dots (CDs) has emerged in recent years as one of the most promising branches in the field of carbon nanomaterials. With outstanding optical properties, these materials have found a wide range of applications as biosensors and fluorescent probes, among many others [1,2]. Graphene quantum dots (GQDs) are considered an important sub-group of the CDs family, exhibiting in general morphological features typical of graphenic materials; such as the existence of basic structural units consisting of nanosized disk-shaped aggregates of stacked graphene-like layers [3,4]. The GQDs combine many beneficial properties typical of graphenic materials (such as chemical stability, biocompatibility and ease of preparation from low-cost precursors) with remarkable photoluminescent features [3,4]. GQDs are well-known for their low cytotoxicity [3] and for being promising candidates in applications such as bioimaging, optoelectronics, electrochemical energy storage, sensing and catalysis, among others [4]. The use of GQDs as fluorescent probes for cell imaging is particularly promising, as demonstrated in several recent reports [5–8].

GQDs can be obtained following different routes, broadly divided in two classes: top-down and bottom-up approaches. In the bottom-up methods, solution-based chemical routes allow the synthesis of GQDs starting from several types of organic precursors; such as malic acid, urea and other organic precursors [4,7,8]. In these cases, the size and colloidal stability of the GQD dispersions can be controlled; however, the yield is generally low and purification treatments are required in order to obtain the final product [5]. The

top-down methods involve the size reduction of a carbon-rich precursor, leading to the achievement of small graphenic aggregates (nanodots) [2–4,9]. Several routes have been proposed to achieve this end, such as oxidative cutting [10], hydrothermal/solvothermal methods [5,6,11,12] and electrochemical processes [13], among others. Numerous precursors have also been used in these syntheses, such as graphite [9,12], graphene oxide (GO) [11,13–15], carbon fibers [16] and carbon nanotubes [12,15]. Nevertheless, the production of GQDs at large scale, with uniform size, high yield and low cost, remains a challenge.

This work involves the use of electronic waste (i.e., graphite recycled from the anode of spent Li-ion batteries) as the starting point to obtain technologically important materials such as the GQDs, following a simple and low-cost method. GO was prepared from the recycled graphite via a modified Hummers method; alcoholic aqueous GO suspensions (with no addition of acids or strong oxidizing agents) were submitted to hydrothermal treatments at different temperatures using a homemade hydrothermal reactor (constructed from recycled pieces of old instruments) that allows the control of the heating program and the assessment of the internal pressure generated in the reaction. The synthesized GQDs exhibited bright blue/green luminescence under UV light, showing the success of the chosen route and opening the way for future applications of these materials in the field of optoelectronic devices.

## 2. Materials and Methods

### 2.1. Samples Preparation

GO was prepared by a modified Hummers method [17,18], using recycled graphite derived from spent lithium-ion batteries (LIBs) [19]. The spent LIBs were first dismantled into their different parts (including cathode, anode, organic separator, plastic and metallic shell). The graphite in the anode was separated from the copper foil by dissolution in concentrated nitric acid solution; followed by filtration, washing with distilled water and drying at ca. 100 °C for 24 h. For the graphite oxidation reaction, about 1.0 g of recycled graphite was added to 70 mL of $H_2SO_4$; followed by the slow addition of 3.0 g of $KMnO_4$ and 0.5 g of $NaNO_3$. The mixture was stirred for 2 h in an ice bath. After that, 400 mL of water and 10 mL of $H_2O_2$ (35%) were added to the reaction medium. The obtained product was washed with distilled water until the filtrate reached a pH next to 5–6 and then dried at 50 °C for 24 h. After drying, the obtained GO thick film was manually cut into small flakes; then, a suspension was prepared using 100 mg of GO added to 50 mL of 99.8% ethanol and 50 mL of distilled water. Afterwards, this suspension was sonicated during 40 min to obtain a homogeneous dispersion.

The synthesis of the GQDs followed a hydrothermal route similar to that described elsewhere [14,15]. It is worth emphasizing, though, that in this work no acids or strong oxidizing agents (e.g., $H_2O_2$) were added to the suspension submitted to the hydrothermal treatments. The alcoholic aqueous GO suspension was transferred to a Teflon-coated stainless steel homemade hydrothermal reactor and thermally treated at programmed temperatures of 125 and 175 °C during 2 h. The hydrothermal reactor used in this work (illustrated in Figure 1) is a homemade apparatus constructed from recycled pieces of old instruments. The main part is a stainless-steel vessel (which was originally part of a bomb calorimeter) containing a Teflon cup fitted to its internal volume. The two openings on the top part are used to insert the temperature sensor (K-type thermocouple) and the pressure transducer (comprising a flexible tube, such as in Bourdon manometers). The pressure is measured with the help of a Hall probe and a permanent magnet; the voltage change detected by the Hall probe when displaced with respect to the magnet by the deformation of the flexible tube is externally calibrated to allow the determination of the internal pressure in the vessel. The system is heated using an electrical band heater, containing electrical resistances with mica insulation and a stainless-steel shield. The computer control of the heating program and the pressure monitoring are implemented using an Arduino prototyping platform [20], as a microcontroller, coupled to a LabVIEW graphical interface [21]. The heating rate, final temperature and residence time are adjusted by the

user via a LabVIEW proportional-integral-derivative (PID) controller that sets the electrical power to be delivered; activating a TRIAC as a power switch via the Arduino board. The communication between the Arduino board and the LabVIEW interface is provided by a homemade program written in C (which is freely available upon request). With this system, it is possible not only to set the operational temperature and the residence time, but also to record in real time the pressure and the temperature of the reaction medium inside the reactor. The temperature and pressure data as a function of time are collected by the Arduino board and sent to the LabVIEW interface for graphical representation in the computer screen; in addition, they are also saved in a log file for posterior data processing.

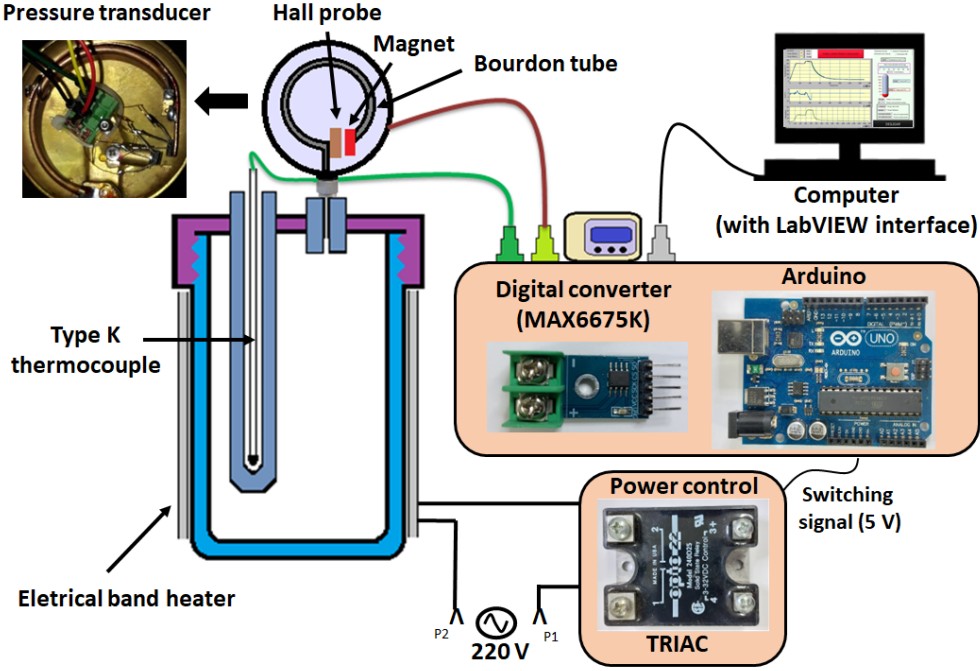

**Figure 1.** Scheme illustrating the operation of the homemade hydrothermal reactor.

After the conclusion of the reaction, the samples were left to cool down to room temperature naturally for 12 h. Afterwards, the obtained suspensions were centrifuged at 4000 rpm for 3 h; then, the supernatant was carefully collected and left to dry in a stove at 50 °C for 24 h. The dry material was then redispersed in 50 mL of distilled water using an ultrasonic bath for 20 min. The dispersion was double filtered to remove large particles/aggregates still present in the material and finally, the filtered dispersion was characterized by the methods described below. These samples were labeled by their temperature of synthesis; i.e., GQD_125 and GQD_175 are the labels of the samples synthesized at 125 and 175 °C, respectively. For comparison, a reference sample was also prepared by simply adding 2 mg of GO to 100 mL of distilled water (i.e., with no hydrothermal treatment); this suspension was sonicated for 40 min and filtered afterwards to remove the undispersed material. This product was labeled as GO_Ref.

*2.2. Characterization*

The produced GO sample was characterized by thermogravimetry (TG), X-ray diffraction (XRD) and solid-state nuclear magnetic resonance (NMR) spectroscopy. The TG curve was recorded using a TGA-50H Shimadzu thermobalance, with $O_2$ flow (50 mL/min) and a heating rate of 2 °C/min; the sample was mixed to alumina powder in order to avoid the loss of material following the strong combustion reaction. The XRD pattern was recorded in a XRD-6000 Shimadzu diffractometer, using Cu-K$\alpha$ radiation ($\lambda$ = 1.5418 Å); and with the diffraction angle 2$\theta$ ranging from 5 to 50° in 0.04° steps. The solid-state $^{13}$C NMR spectrum was recorded at room temperature in a Varian/Agilent VNMR 400 MHz spectrometer

(NMR frequency of 100.52 MHz, magnetic field of 9.4 T), with the powdered sample packed into 4 mm diameter zirconia rotors for magic angle spinning (MAS) experiment at the spinning rate of 14 kHz; the spectrum was recorded using direct excitation of the $^{13}$C nuclei ($\pi/2$ pulse of 4.3 µs), with a recycle delay of 15 s and accumulation of ca. 4000 transients. The chemical shifts were referenced to tetramethylsilane (TMS), using hexamethylbenzene as a secondary reference (signal at 17.3 ppm).

The produced GQDs and the GO_Ref sample were first analyzed by recording pictures of the aqueous suspensions in the dark under a 5 mW ultraviolet (UV) light with a wavelength of 365 nm produced by a commercial UV LED flashlight (Nitecore, model GEM 10 UV). The UV–visible (UV–Vis) optical absorption of the samples was analyzed using a Globlal Analyzer GTA-97 spectrophotometer. Fluorescence spectra were recorded in a Perkin Elmer LS 55 Fluorescence spectrometer. Finally, transmission electron microscopy (TEM) images were recorded using a JEOL microscope, model JEM-1400.

## 3. Results and Discussion

### 3.1. Chemical and Structural Characterization of GO

The TG curve obtained for the synthesized GO sample is shown in Figure 2. The weight losses typical of well-oxidized GO samples are clearly identified in this curve, including the water release from room temperature up to ~150 °C; the decomposition of oxygenated functions (such as epoxy, hydroxyl and carbonyl groups) between ca. 150 and 350 °C; and the oxidation of the remaining graphitic structure between ca. 400 and 700 °C [18,22]. The GO ash content (estimated from the final mass in the TG curve shown in Figure 2) is around 2.0 wt.%; which is attributed to the presence of metal impurities coming from the recycled graphite and/or introduced during the graphite oxidation reaction.

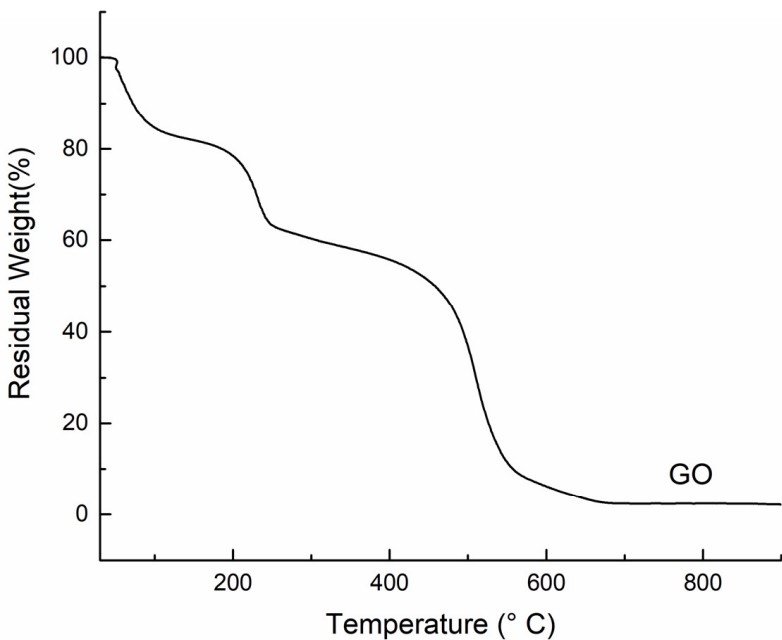

**Figure 2.** TG curve recorded under $O_2$ flow for the synthesized GO.

The well-oxidized nature of this GO sample is also revealed in the XRD pattern shown Figure 3, which is dominated by a peak at approximately 12°. The low value of this Bragg angle is associated with an increased interlayer spacing (around 7 Å) in comparison with graphite (3.35 Å); this is a consequence of the presence of oxygen-containing groups and intercalated water molecules in the GO structure [18,22,23].

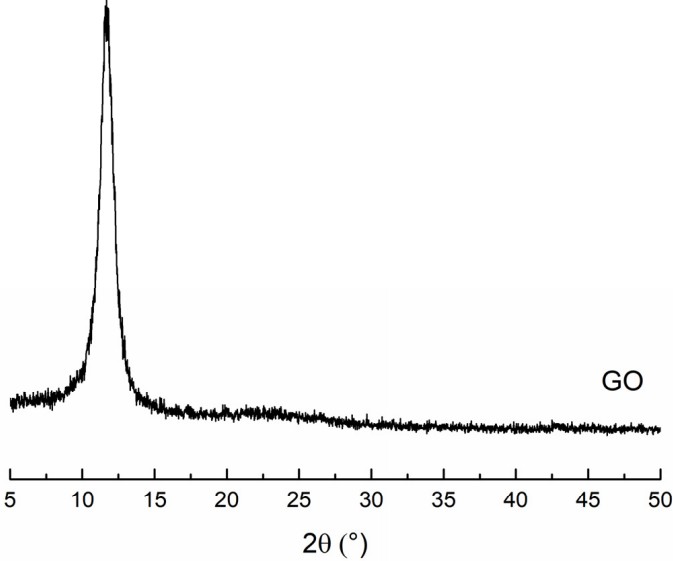

**Figure 3.** XRD pattern of the synthesized GO.

The nature of the oxygen-containing functional groups present in the GO structure was elucidated using solid-state [13]C NMR spectroscopy; the spectrum obtained for the synthesized GO sample is exhibited in Figure 4. The most characteristic [13]C NMR signals due to the oxygen-containing groups are observed at 70 ppm (C-OH groups) and 60 ppm (epoxy groups). In addition to these, the strong signal close to 130 ppm, along with the corresponding spinning sidebands at ca. 270 and −10 ppm, are assigned to $sp^2$-hybridized carbon atoms in hexagonal rings. A number of other weaker signals can also be observed, such as the ones associated with lactol (100 ppm), carbonyl (190 ppm) and carboxyl (167 ppm) groups [18,24,25].

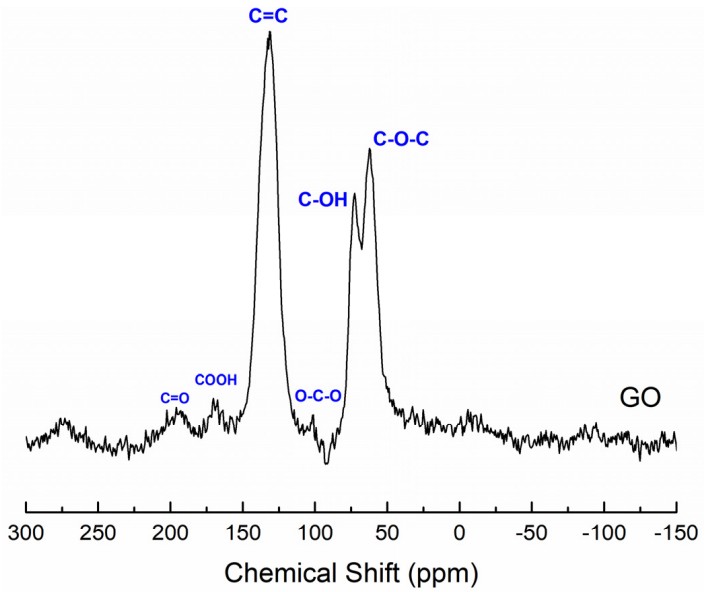

**Figure 4.** Solid-state [13]C NMR spectrum obtained for the GO sample.

Therefore, this set of results of chemical/structural characterization revealed that the synthesized GO sample exhibited indeed a well-oxidized structure, with plenty of oxygen-containing functional groups; thus, it is suitable to be used in the hydrothermal treatments to obtain the GQDs, as described below.

### 3.2. Hydrothermal Treatments

The apparatus used in this work revealed interesting findings on how the temperature and the pressure of the reaction medium inside the hydrothermal reactor evolved over time during the synthesis of the GQDs; this is shown in Figure 5. Regarding the temperature profile, it is possible to observe an overshoot of ~15–20 °C in the beginning of the temperature plateau (Figure 5a,c), even after careful adjustment of the temperature control parameters. A similar pattern was observed for the internal pressure (Figure 5b,d), which showed a significant increase in the beginning of the temperature plateau, before reaching a nearly stable value. The internal pressure fluctuations in this plateau closely match the temperature fluctuations, as expected. The maximum pressure reached values of ca. 3 and 23 bar; whereas the final values were around 0.5 and 11 bar, for the hydrothermal treatments performed at 125 and 175 °C, respectively.

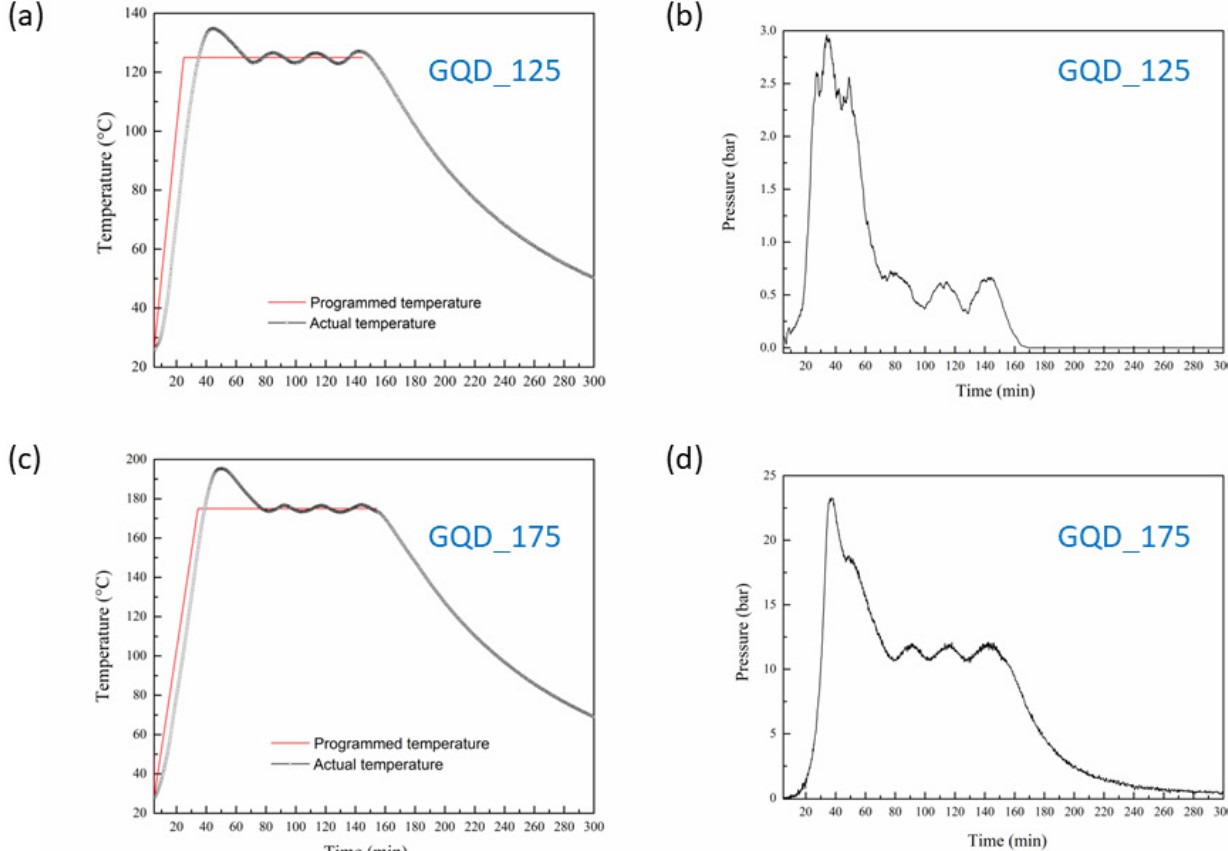

**Figure 5.** Evolution of temperature (parts (**a**,**c**)) and pressure (parts (**b**,**d**)) inside the hydrothermal reactor during the syntheses of the GQD_125 (parts (**a**,**b**)) and GQD_175 (parts (**c**,**d**)) samples.

### 3.3. Optical and Structural Properties of the GQDs

Figure 6 shows the pictures of the aqueous suspensions of the GQDs taken under 5mW, 365 nm UV light; these images exhibit clearly a bright bluish-green glow for both GQD samples (Figure 6b,d). For comparison, Figure 6f shows the corresponding picture obtained for the reference GO suspension (GO_Ref), where no glow is observed; pointing to the absence of quantum dots in this sample, as expected. On the other hand, all the samples (GO_Ref, GQD_125 and GQD_175) looked perfectly transparent under white light (Figure 6a,c,e). These results thus indicate that both produced GQD samples exhibit photoluminescent behavior when irradiated under UV light, which is a typical feature of aqueous GQD suspensions [3–5,9].

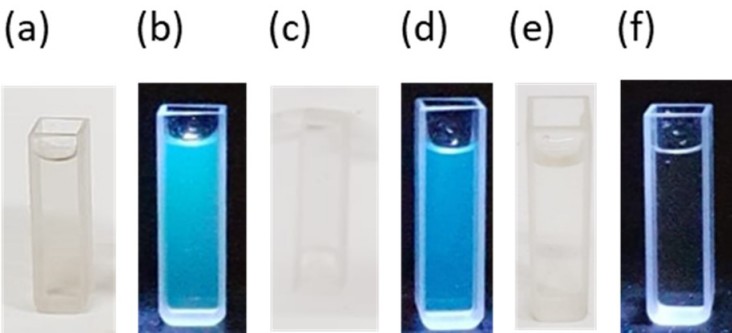

**Figure 6.** Images of aqueous suspensions of samples GQD_125 (parts (**a**,**b**)) and GQD_175 (parts (**c**,**d**)); and sample GO_Ref (parts (**e**,**f**)). The pictures shown in parts (**a**,**c**,**e**) were recorded under white light; whereas the ones in parts (**b**,**d**,**f**) were recorded under UV light.

The UV–Vis absorption spectra obtained for the produced samples are shown in Figure 7a; these spectra show a strong absorption band centered close to 225 nm, as usually observed in GO and undoped GQDs dispersions [9,14,15]. Even though the reference GO suspension (GO_Ref sample) also exhibits an absorption band in this region, it is important to highlight that it is not as intense as for the GQDs dispersions [14]. The fluorescence spectra recorded in the visible spectral region under 365 nm excitation for the produced samples are shown in Figure 7b; these spectra confirm the photoluminescent behavior qualitatively observed in the images shown in Figure 6. As expected, no fluorescence was detected for the GO_Ref sample, whereas the fluorescence signal was clearly observed with maximum intensity around 450 nm for both GQDs dispersions; this is in good agreement with previous reports in the literature [14,15].

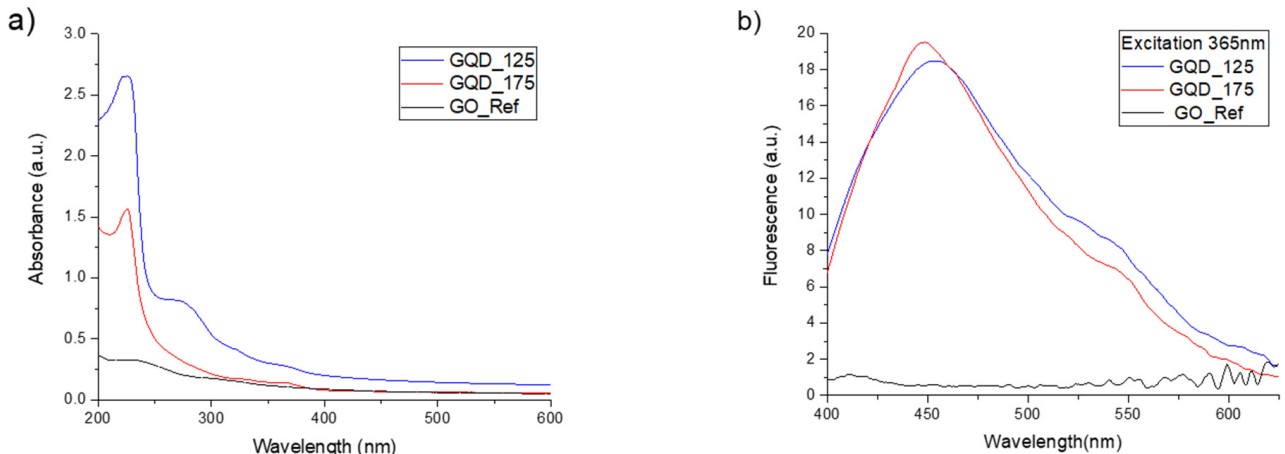

**Figure 7.** (**a**) UV–Vis absorption spectra and (**b**) fluorescence spectra (under 365 nm excitation) obtained for aqueous suspensions of the GQD_125, GQD_175 and GO_Ref samples.

Finally, the TEM images recorded for the produced samples (Figure 8) reveal that the GQD_125 and GQD_175 samples are composed of particles with average sizes of around 60 and 30 nm, respectively. The trend of smaller particles being produced at higher treatment temperatures is in agreement with previous reports [14,15]; as is the slight shift in the maximum of the fluorescence spectra obtained for these two samples (Figure 7), with the maximum fluorescence being observed at a slightly lower wavelength for the GQD_175 sample in comparison with the GQD_125 sample.

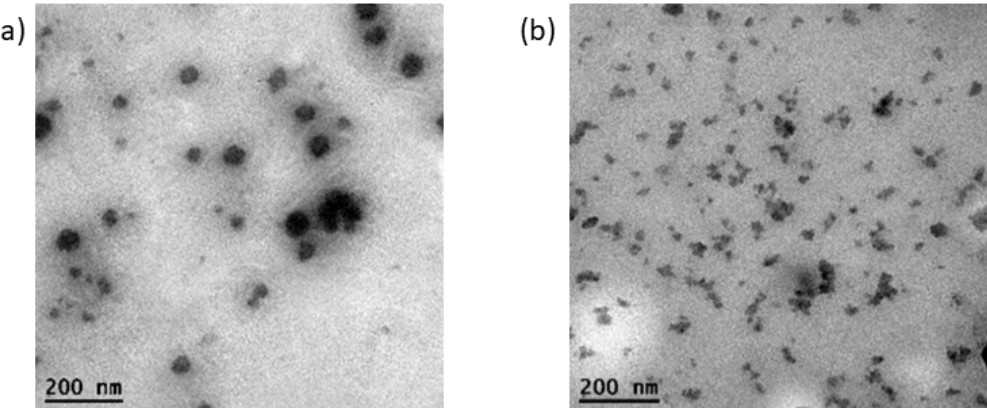

**Figure 8.** TEM images obtained for the GQD_125 (**a**) and GQD_175 (**b**) samples.

## 4. Conclusions

A simple and low-cost route for the synthesis of GQDs, starting from an alcoholic aqueous suspension of graphene oxide derived from recycled graphite and using a home-made hydrothermal reactor, was described in this work. The developed apparatus allowed the control of the heating program and the assessment of the internal pressure generated during the reaction at high temperatures. The synthesized GQDs were fully characterized, with the observation of bright blue/green luminescence under UV light, thus showing the success of the chosen route and opening the way for future applications of these materials in the field of optoelectronic devices.

**Author Contributions:** Conceptualization, J.C.C.F., M.A.S.J.; Data curation, L.M.D., T.L.G.C. and D.F.C.; Formal analysis, J.C.C.F.; Investigation, L.M.D., T.L.G.C., D.F.C., C.W.C. and M.A.S.J.; Methodology, L.M.D., T.L.G.C., C.W.C. and M.A.S.J.; Resources, J.C.C.F.; Writing—original draft, L.M.D. and M.A.S.J.; Writing—review & editing, J.C.C.F. All authors have read and agreed to the published version of the manuscript.

**Funding:** The authors acknowledge the financial support from the Brazilian agencies FAPES (grants 345/2019, 280/2021 and 495/2021), CAPES and CNPq. The authors are also grateful to the Laboratory of Cellular Ultrastructure Carlos Alberto Redins (LUCCAR) and the Laboratory for Research and Development of Methodologies for Crude Oil Analysis (LabPetro), both at the Federal University of Espírito Santo (UFES), for the use of their experimental facilities.

**Data Availability Statement:** The data are available upon request from the authors.

**Conflicts of Interest:** The authors declare no conflict of interest.

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
