# Peer review of "Synthesis of Graphene Quantum Dots by a Simple Hydrothermal Route Using Graphite Recycled from Spent Li-Ion Batteries"

_carbon, 2022_

Round 1

Reviewer 1 Report

This manuscript mainly reports a method to prepare graphene quantum dots (GQDs) using graphene oxide (GO) as the precursor from Li-ion batteries. It belongs to waste recycling, but the paper has no further application, which is obviously not enough. I suggest major revisions because the lack of application. In the introduction, the author emphasizes the importance and great application value of GQDs, but the manuscript lacks it.

1. The author added acid-free to the title. But, there is strong acid in the process of obtaining graphene oxide, which is contradictory, please revise.

2. Why should oxygen be passed through the TG process? What is it like when there is no oxygen? Please compare them.

3. About the application, provide you several references: Molecules. 2021, 26, 4994. Chemical Engineering Journal. 2016, 300, 7582.

4. In the introduction, GQDs are the focus of this manuscript, which should be introduced mainly, including its preparation method, fluorescence, application, etc. The general morphology and structure characterization of GQDs are lacking. Such as atomic force microscopy (AFM), Transmission Electron Microscope (TEM), Raman, X-ray photoelectron spectroscopy (XPS). Optical performance test: excitation dependence, emission spectrum and pH tolerance. Related characterization can refer to the literature: Nanoscale Horiz. 2020, 5, 928–933. Science Advances. 2020, 6, eabb6772.

Reviewer 2 Report

The manuscript submitted by Darabian et al. reports on the synthesis of graphene nanoplatelets from spent batteries, using its graphite anode. The methodology is partly new and partly old: the synthesis is based on the traditional Hummers method and in that respect it is rather routine. However, the idea of using wastes are great, it is always important in terms of environmentally friendliness. It is interesting to remove all possible contaminants from the spent battery-graphite and the authors did efforts to do that, but it would be intersting to know, in a later study, if the copper ions remained adsorbed in the graphite after the purification step. Likely not, but if there are already some oxidized carbon speciss in graphite, divalent copper ions may be retained and alter the hyddothermal process. 

I think that the manuscript deserves publication because the reported properties of the GNPs are good in terms of fluorescence properties. However, the authos should explicitly write in the introduction, what is exactly new: they state that the hydrothermal protocol has been used before, and citations no 10 and 11 also report on the hydrothermal synthesis of GQD-s...in that respect, the novely is missing. Was the composition of the medium new, or the employed temperature range?

Please correct "reactional" medium to reaction medium.

question: why was it needed to double-filter the obtained suspension, once it was centrifuged and only the supernatant was used for futher steps? How could any "contaminant" get into the sample that needed to be removed in the forthcoming steps?

Reviewer 3 Report

The Article “Synthesis of Graphene Quantum Dots by Acid-Free Hydrothermal Route Using Graphite Recycled from Spent Li-ion Batteries” is written nicely. The application of using graphite recycled from spent Li-ion batteries is unconventional. For clarity, after reading the preparative steps, I am convinced that the end product can be contaminated with Cu, Li, and Mn ions. Rinsing with distilled water does not purify the system too well small addition of 1% of HCl improves it, but contaminations will be there even so. Nevertheless, the characterization methods used did not detect any additional signals, so the material is clean for these methods. The authors used standard characterization techniques, and if not, the control of the system by Arduino novelty of this work would be low.

I propose to the authors to better describe the Arduino part in the text. Please improve the graphs mentioned below, and answer the questions in the main text.  

Fig 1 Is there a feedback signal going to the heater to set the temperature? If yes, an additional cable should go from Arduino to the heater controller or from the computer to the heater. Controls for each sensor are drawn together with Arduino. Could you magnify this part so that it is more informative and write the commercial names of the controllers? Please provide more details about these connections. I am curious.

Can you overwrite the Arduino internal program from LabView or only read the sensor data?

Fig 2 what is the final mass of the sample in this plot?

Fig 4 With which substance the chemical shift was calibrated?

Fig 5 Where are the pressure fluctuations inside the autoclave coming from?

Round 2

Reviewer 1 Report

The author has improved related issues and can be accepted.

Reviewer 3 Report

The authors improved the manuscript greatly (images and text) and answered all the questions raised before. I recommend for publishing.